# Interventions to reduce opioid use for patients with chronic non-cancer pain in primary care settings: A systematic review and meta-analysis

Qian Cai[1]*, Christos Grigoroglou[2], Thomas Allen[2], Teng-Chou Chen[3], Li-Chia Chen[3], Evangelos Kontopantelis[1,4]

1 Division of Informatics, Imaging & Data Sciences, School of Health Sciences, Faculty of Biology, Medicine and Health, Manchester Academic Health Science Centre, University of Manchester, Manchester, the United Kingdom, 2 Centre for Health Economics, Division of Population Health, Health Services Research and Primary Care, School of Health Sciences, Faculty of Biology, Medicine and Health, Manchester Academic Health Science Centre, University of Manchester, Manchester, the United Kingdom, 3 Centre for Pharmacoepidemiology and Drug Safety, Division of Pharmacy and Optometry, School of Health Sciences, Faculty of Biology, Medicine and Health, University of Manchester, Manchester, the United Kingdom, 4 National Institute for Health and Care Research (NIHR) School for Primary Care Research, Division of Population Health, Health Services Research and Primary Care, School of Health Sciences, Faculty of Biology, Medicine and Health, Manchester Academic Health Science Centre, University of Manchester, Manchester, the United Kingdom

* qian.cai@manchester.ac.uk

## Abstract

### Objective

This systematic review and meta-analysis aimed to assess interventions to reduce opioid use for patients with chronic non-cancer pain (CNCP) versus usual care or active controls in primary care settings.

### Methods

In this registered study (PROSPERO: CRD42022338458), we searched MEDLINE, Embase PsycInfo, CINAHL, and Cochrane Library from inception to December 28th 2021, and updated on Dec 14th 2023 for randomized controlled trials (RCTs) and cohort studies with no restrictions. Methodological quality was assessed using the Cochrane Risk of Bias tool for RCTs and Newcastle Ottawa Scale for cohort studies. Primary outcomes included mean reduction in morphine equivalent daily dose (reported as mean differences [MDs] mg/day; 95% confidence intervals [95%CIs]) and/or opioid cessation proportion. Secondary outcomes were mean changes in pain severity (reported as standardized mean difference [SMDs]; 95%CIs) and (serious) adverse events. Meta-analyses were performed using random-effects models.

### Results

We identified 3,826 records, of which five RCTs (953 participants) and five cohort studies (901 participants) were included. Overall, opioid dosage was significantly reduced in intervention groups compared to controls (MD: -28.63 mg/day, 95%CI: -39.77 to -17.49; $I^2$ =

**Funding:** The author(s) received no specific funding for this work.

**Competing interests:** The authors have declared that no competing interests exist.

31.25%; eight studies). Subgroup analyses revealed significant opioid dose reductions with mindfulness (MD: -29.36 mg/day 95%CI: -40.55 to -18.17; $I^2$ = 0.00%; two trials) and CBT-based multimodalities (MD: -41.68 mg/day; 95%CI: -58.47 to -24.89; $I^2$ = 0.00%; two cohort studies), respectively, compared to usual care. No significant differences were observed in opioid cessation (Odds ratio: 1.10, 95%CI: -0.48 to 2.67, $I^2$ = 58.59%; two trials) or pain severity (SMD: -0.13, 95%CI: -0.37 to 0.11; $I^2$ = 33.51%; three trials). Adverse events were infrequently examined, with withdrawal symptoms commonly reported.

## Conclusions

The studied interventions were effective in reducing opioid dosage for people with CNCP in primary care. They highlighted the importance of multidisciplinary collaboration. Large-scale RCTs measuring the long-term effects and cost of these interventions are needed before their implementation.

## Introduction

Opioids are primarily recommended by the World Health Organization pain ladder in treating acute pain, cancer pain, and palliative care [1]. During the last two decades, there has been a marked increase in opioid prescriptions issued for chronic non-cancer pain (CNCP), predominantly in the United States (US) [2], Canada [3], and several European countries [4–6] including the United Kingdom (UK) [7]. Despite the widespread utilization of opioids, findings from randomized controlled trials and systematic reviews suggest limited efficacy of these medications, yielding only modest effects in pain relief in the short to medium term (less than 12 weeks) [8]. While the evidence supporting the long-term use of opioids remains sparse, the associated harms of long-term opioid treatment (LTOT), including respiratory depression, bone fractures, and opioid-related mortality, are well documented [9]. Moreover, prolonged opioid use introduces risks of dependence, addiction, and abuse [10].

In response to these concerns, clinical guidelines [11–13] have prompted healthcare providers (HCPs) to reassess their prescribing practices, emphasizing the need to reduce opioid use when potential risks outweigh perceived benefits. However, the endeavor to reduce opioid use encounters significant challenges, including patients' fear of withdrawal symptoms, inadequate social and healthcare support, and limited availability of non-opioid methods for pain management [14, 15]. Currently, there is a lack of guidance and practical support to implement the reduction of LTOT. Therefore, there is an urgent need for an evidence-based evaluation of interventions to assess their effectiveness in reducing opioid utilization and evaluate their impact on clinical outcomes, particularly in primary care settings, where opioids are mostly prescribed.

Previous systematic reviews [16–19] have provided valuable insights into interventions aimed at reducing opioid use for chronic pain patients. However, their inclusion criteria were broad, limiting their relevance to primary care settings. For example, these reviews analyzed interventions (e.g., spinal cord stimulation) that are not readily accessible in primary care settings, and they included studies that evaluated abrupt or gradual opioid tapering protocols without incorporating additional supportive therapies for patients. The provision of supplementary interventions is crucial, as patients often express reluctance to reduce or cease opioids when alternative treatments are not provided, given these medications constitute their primary method of pain management in real-world practices. Furthermore, the absence of additional

interventions impedes the identification of the key components contributing to the dose reduction outcomes. Some prior systematic reviews also included potentially relevant studies. However, those studies focused on managing related symptoms, such as withdrawal symptoms [20], opioid overuse headache [21], or improving adherence to antidepressants to increase the likelihood of opioid cessation [22], rather than directly reducing opioid use, despite a decrease in opioid dosage or an improvement in pain severity was achieved. In view of these limitations and the emergence of new studies recently, there is a compelling need for an updated evaluation of the current evidence.

The present systematic review and meta-analysis aimed to evaluate the effectiveness of healthcare provider-directed interventions designed to reduce or discontinue opioid use for patients with CNCP, with a particular focus on primary care settings. Specifically, the objectives included comparing changes in morphine equivalent daily dose (MEDD), evaluating the proportion of patients discontinuing opioids, and assessing the change in pain severity and adverse events between interventions and usual care or active controls.

## Methods

This registered systematic review and meta-analysis (PROSPERO: CRD42022338458) was conducted following the Cochrane Handbook [23] and the Preferred Reporting Items for Systematic Reviews and Meta-Analyses (PRISMA) guidelines (S1 Appendix) [24].

### Eligibility criteria

Eligible studies were full-scale randomized controlled trials (RCTs) and cohort studies that examined primary care provider-directed interventions, designed to reduce or discontinue opioids in adult patients ($\geq$18 years) with CNCP (persists for $\geq$3 months). Studies exclusively focused on acute pain, cancer pain, surgical pain or palliative care were excluded. Pregnant or breastfeeding women, non-human participants and patients with substance use disorders (e.g. opioid use disorders) were not our target study population and were thus excluded. Studies that concurrently used opioids with other analgesics (e.g., non-steroidal anti-inflammatory drugs) were excluded, as distinguishing the analgesic effects of opioids from those analgesics were challenging. Interventions implemented in hospital settings or managed by patient themselves were not considered for inclusion. Studies that solely focused on reducing opioids without offering patients alternate treatments or replacements were excluded. Interventions were considered if they explicitly aiming to reduce or cease opioid use, whereas those with a spillover effect on opioid use were excluded. Any comparator was accepted, including usual care or active controls (either pharmacological or non-pharmacological).

The primary outcomes of interest included: 1) opioid dose reduction, measured by the mean changes in morphine milligram equivalent daily dose (MEDD) from pre- to post-treatment. The homogeneity of this outcome measure allowed us to examine our study objectives meta-analytically; 2) The proportion of participants for whom opioid use was either ceased or declined. Secondary outcomes included the mean change in pain severity (measured by pain rating scales on a range of 0–10, where 0 indicating no pain whilst 10 indicating severe pain) and the number of (serious) adverse events related to opioid reduction. Case reports, cross-sectional studies, case-control studies, pilot studies, reviews or meta-analyses were excluded.

### Data sources and search strategies

We performed comprehensive searches in databases including MEDLINE, EMBASE, PsycINFO, CINAHL, and Cochrane Central Register of Controlled Trials. Searches were conducted from the inception of each database until December 28th 2021, and updated on

December 14[th] 2023, using structured search strategies (S2 Appendix) that incorporated text words and medical subject headings related to "chronic pain", "opioids", and "reduce/discontinue/cease/deprescribe/". No language and geographic restrictions were applied. Ongoing trials or unpublished studies were obtained from ClinicalTrials.gov. To ensure literature saturation, we manually retrieved additional studies from the reference lists of included studies and published systematic reviews.

## Study selection

Two review authors (QC and CG) independently screened titles and abstracts of the retrieved citations against pre-determined inclusion/exclusion criteria. When necessary, full text of eligible records were reviewed for further eligibility assessment. Where discrepancies occurred and remained unresolved by discussion, a third party (EK, TA and LCC) was consulted for adjudication. The inter-rater agreement test demonstrated a high level of consistency (99.64%) between QC and CG with a Kappa coefficient equals to 0.7982 (p<0.001).

## Data extraction and quality assessment

Data extraction was undertaken independently by one review author (QC) and verified by another reviewer (TCC) for accuracy. The study authors were also contacted by email to request any necessary missing information. The following key information was extracted:

- Study: first author, publication year, country, study design, settings.

- Patient characteristics: age, gender, sample size, pain duration and severity, opioid use status, comorbidities.

- Intervention and comparator: key components, mode of administration, frequency, treatment duration.

- Outcome: opioid dose, pain severity, adverse events.

## Risk of bias assessment

The methodological quality of all included studies was independently appraised by QC. For RCTs, the Cohrane's Risk of Bias Tool 2.0 (RoB 2) [25] was employed to assess the risk of bias, including 1. sequence generation; 2. allocation concealment; 3. masking of participants, staff and outcome assessors; 4. incomplete outcome data; and 5. selective outcome reporting. For cohort studies, a modified Newcastle-Ottawa Scale (NOS) [26] was used to evaluate bias risk, focusing on 1. the representativeness of the exposed cohort; 2. selection of the non-exposed cohort; 3. ascertainment of exposure; 4. absence of the outcome of interest at the start of the study; 5. comparable controls; 6. assessment of outcome; 7. sufficient follow-up length for outcomes to occur; and 8. adequacy of follow-up cohorts. For RCTs, high quality was defined as minimum 4 domains of low risk. As for cohort studies, the modified NOS adopted a star rating system. A study was deemed to be of good quality if it obtained ≥3 stars in selection domain (1–4), ≥1 star in comparability domain (5), and ≥2 stars in outcome/exposure domain (6–8).

## Data synthesis and analysis

The study characteristics and details of opioid reduction interventions were presented descriptively. Effect sizes were reported using odds ratios (ORs) with their 95% confidence intervals (95%CIs) for opioid cessation, mean differences (MDs) with 95%CIs for opioid dose reduction, and standardized mean differences (SMDs) with 95%CIs for pain severity, given the

utilization of different scales for this outcome measure. Magnitude of effect was defined as large (SMD > 0.8), medium (SMD 0.5–0.8), small (SMD 0.2–0.5) or trivial (SMD < 0.2) [27].

Where between- or within-group SDs were not reported, relevant data such as sample size, p values, t statistics, standard errors (SEs), or 95% CIs were used to derive SDs using the formula recommended by the Cochrane Handbook (6.5.2.3) [23] or the calculator embedded in Review Manager 5.4. Additionally, when outcomes were assessed at multiple time points during long-term follow-up, data from the last available time point were employed.

Heterogeneity was assessed using the $I^2$ statistics, with values below 50% suggesting low heterogeneity, 50–75% moderate, above 75% substantial heterogeneity [28]. Random effects meta-analyses with a non-parametric bootstrap of DerSimonian Laird (DL) method were conducted for pooling the outcomes of interest [29, 30] even if $I^2$ was low. Publication bias regarding the primary outcomes was not assessed due to the small number of included studies.

Sensitivity analyses were conducted by sequentially removing one study at a time and repeating the meta-analysis based on the remaining data to assess whether pooled estimates were unduly influenced by specific studies. Furthermore, multivariable random effects meta-regressions were conducted to explore potential sources of heterogeneity, including patient mean age; gender; study period; sample size; follow-up time point; intervention type and category. Subgroup analyses were undertaken based on intervention types, which included mindfulness techniques, and CBT-based multi-component strategies. All data analyses were conducted with Stata/MP 17.0.

## Results

### Study selection and patient characteristics

The initial search retrieved 3,826 potentially relevant records, of which 1,126 duplicates were removed, yielding 2,770 unique records for eligibility assessment. By screening titles and abstracts against our predefined inclusion/exclusion criteria, 2740 studies were excluded. Of the remaining 30 records, ten full-text articles including five RCTs (953 participants, female 557 [58.4%], mean age: 58.46±2.90) [31–35] and five retrospective cohort studies (patients 901, female 188 [20.9%], mean age: 57.38±7.63) [36–40] published between 2016 and 2023 were included in this systematic review and meta-analysis (Fig 1). All included studies reported the baseline opioid dosage (intervention group: mean 84.70±51.72 mg/day vs comparators: mean 100.63±64.24 mg/day), with three [34, 37, 39] indicating that participants consumed a mean MEDD of >120 mg/day. None of the studies specified the exact CNCP conditions. Studies were mainly conducted in the US (n = 10), with one in the UK [33]. Both RCTs and retrospective cohort studies had a small sample size ranging between 35 and 608 (Table 1).

### Characteristics of interventions

The components, duration, frequency, and delivery mode of the interventions varied significantly across studies. Two studies focused on physical interventions, incorporating components such as yoga, Taichi, chiropractic therapy [40] and acupuncture [36]; Five studies involved psychological or behavioral changes, integrating key components such as cognitive behavior therapy (CBT), mindfulness techniques, patient education and pain coping skills training [31–33, 35, 39]; Three studies [34, 37, 38] employed mixed multimodal approaches delivered by a multidisciplinary team comprising physicians, nurses, psychologists, pharmacists, and social workers. These multimodal care programs typically combined CBT, mindfulness, acupuncture, chiropractic care, exercise, pharmacotherapies are the core components.

The most common comparator was usual care, which unfortunately was not clearly defined in the included studies. Treatment durations ranged from eight weeks to 12 months. Follow-

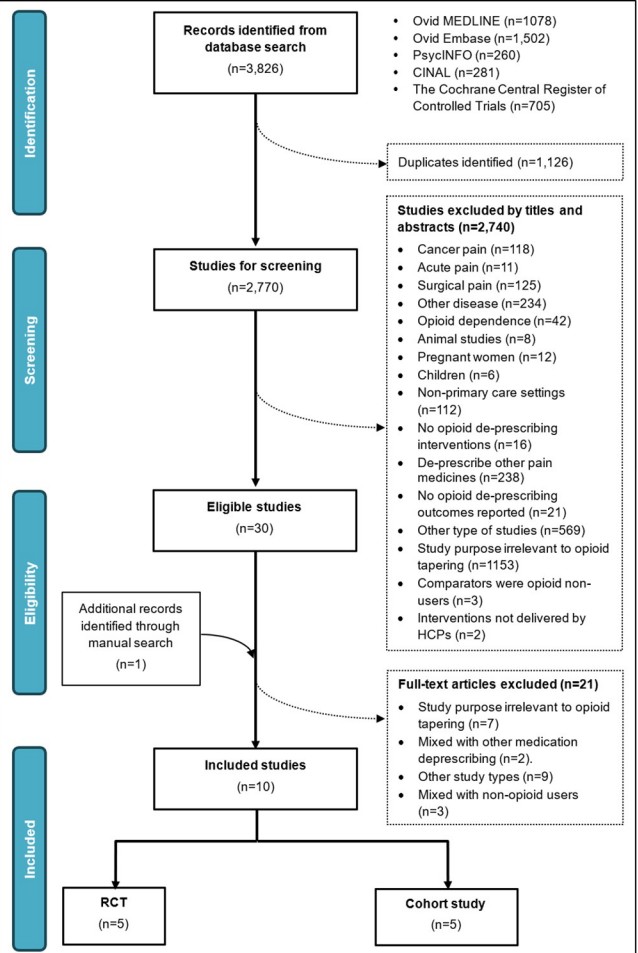

**Fig 1. Selection of included studies.**

up periods (median 6 months) varied from short (≤3 months) [31, 32] to intermediate (6–9 months) [34, 36–40], with three studies reporting long-term outcomes of opioid reduction (≥12 months). (See Table 2 for characteristics of interventions).

## Risk of bias assessment

The Cochrane RoB-2 assessment indicated an overall risk of bias being low (n = 3) to moderate (n = 2) (Fig 2). Due to the impossibility of masking the interventions, participants and researchers were unblinded, which increased risk. The high dropout rate during follow-ups was another main reason contributing to the increased risk of bias. The NOS evaluation tool identified three cohort studies [37, 38, 40] with good quality, one [39] with fair and one [36] with poor quality (Table 3). Uncontrolled confounders and limited representativeness of the study population were the main reasons that reduced the quality of these studies.

## Opioid dose reduction

Nine studies including four RCTs [31, 32, 34, 35] and five cohort studies [36–40] reported the outcomes of opioid dose reduction. Opioid dosage was significantly decreased in the intervention groups compared to controls (MD: -34.82 mg/day, 95%CI: -52.24 to -17.39; nine studies;

**Table 1. Characteristics of included studies.**

| Author, year and country | Study design | Population | interventions | Comparators | Outcomes of interest assessed | Quality |
|---|---|---|---|---|---|---|
| **Randomized controlled trials (n = 5)** | | | | | | |
| Sandhu *et al.* 2023 [33], UK | Non-blind, two-arm, RCT | CNCP (n = 608) female 362 (60%) | Skill-based learning integrating education and mindfulness (n = 305) | Usual care (n = 303) | • Opioid cessation • Pain severity • Adverse events | Good |
| Wartko *et al.* 2023 [35], USA | Non-blind, two-arm, RCT | CNCP (n = 153) female 98 (64%) | CBT-based pain coping skills training, education and usual care (n = 79) | Usual care (n = 74) | • Opioid dose reduction • Pain severity • Adverse events | Good |
| Hudak *et al.* 2021 [32], USA | Non-blind, two-arm, RCT | CNCP (n = 62) female 9 (14.5%) | The MORE protocol: Mindfulness (n = 34) | The supportive care (n = 28) | • Opioid dose reduction | Good |
| Garland *et al.* 2020 [31], USA | Non-blind, two-arm, RCT | CNCP (n = 95) female 63 (66.3%) | The MORE protocol: Mindfulness (n = 50) | The supportive care (n = 45) | • Opioid dose reduction | Fair |
| Sullivan *et al.* 2017 [34], USA | Non-blind, two-arm, RCT | CNCP (n = 35) female 25 (71.4%) | CBT-based multimodality & self-care & pain education (n = 18) | Standard of care (n = 17) | • Opioid dose reduction • Opioid cessation • Pain severity | Fair |
| **Cohort studies (n = 5)** | | | | | | |
| Montgomery *et al.* 2020 [36], USA | Retrospective cohort study | CNCP (n = 47) female 5 (10.6%) | Acupuncture (n = 24) | Standard of care (n = 23) | • Opioid dose reduction • Pain severity | Poor |
| Seal *et al.* 2019 [37], USA | Retrospective cohort study | CNCP (n = 294) female 30 (10.2%) | Integrated multimodality including CBT, mindfulness, pain education, acupuncture and exercise (n = 147) | Standard of care (n = 147) | • Opioid dose reduction | Good |
| Oldfield *et al.* 2018 [38], USA | Retrospective cohort study | CNCP (n = 105) female 7 (6.7%) | Multimodality integrating CBT, exercise, and acupuncture (n = 66) | Standard of care (n = 39) | • Opioid dose reduction | Good |
| Goodman *et al.* 2018 [39], USA | Retrospective cohort study | CNCP (n = 41) female 22 (53.7%) | Opioid reduction program & consultation with GPs (n = 27) | Usual care (n = 14) | • Opioid dose reduction | Fair |
| Mehl-Madrona *et al.* 2016 [40], USA | Retrospective cohort study | CNCP (n = 414) female 124 (59.9%) | Medical care and physical exercise (n = 207) | Standard of care (n = 207) | • Opioid dose reduction | Good |

Note: CNCP = Chronic Non-Cancer Pain, MORE = Mindfulness-Oriented Recovery Enhancement, CBT = Cognitive Behavioral Therapy, GPs = general practitioner

Fig 3). However, considerable heterogeneity was noted ($I^2$ = 83.28%, 95%CI: 57.5% to 93.4%, p = 0.00; Fig 3). A sensitivity analysis was conducted to investigate the source of heterogeneity and the Mehl-Madrone et al. (2016)'s study [40] was identified as an outlier (Fig 4). Upon its removal, the heterogeneity decreased from high ($I^2$ = 83.28%) to low ($I^2$ = 31.25%), but the significance or direction of the pooled effects in our meta-analyses remained unchanged (MD: -28.63 mg; 95%CI: -39.77 to -17.49; eight studies; Fig 5). Additionally, a multivariable meta-regression analysis was conducted, revealing that heterogeneity could be partially explained by differences in the longest follow-up time across studies (p = 0.014).

Within the four RCTs [31, 32, 34, 35] provided data that enabled the pooled calculation of change for opioid reduction among 181 participants receiving different interventions compared to 162 receiving usual care (MD: -24.40 mg; 95%CI: -36.32 to -12.47; $I^2$ = 9.21%; Fig 6), a subgroup meta-analysis of two trials [31, 32] using the MORE protocol (mindfulness) found a significant reduction in opioid dosage in 84 patients receiving this intervention, compared to 77 having supportive care (MD: -29.36 mg; 95%CI: -40.55 to -18.17; $I^2$ = 0.00%; Fig 6). Similar decreases in opioid dose were reported by Wartko et al. (2023) [35] and Sullivan et al. (2017) [34] using CBT-based multi-component interventions. However, no between-group statistical significance was attained (MD: -10.10 mg; 95%CI: -33.61 to 13.40; $I^2$ = 0.00%; Fig 6).

Within the four cohort studies [36–39], where various interventions were implemented, a significant decrease in the use of opioid medications was observed in the intervention groups

**Table 2. The components and implementation procedures of each intervention.**

| Studies | Interventions | Duration | Implemented by |
|---|---|---|---|
| **Physical procedure (n = 2)** | | | |
| Montgomery *et al.* 2020 [36] | Battlefield Acupuncture: 5-point auricular acupuncture procedure implemented | 9 months | Physicians |
| Mehl-Madrona *et al.* 2016 [40] | Twice monthly GMV (12 sessions) & weekly physical activity (yoga, exercise class, chiropractic therapy, osteopathic treatment, tai chi, or qigong) | 6 months | Family doctor, nurse, and behavioral health specialist |
| **Psychological or behavioral intervention (n = 5)** | | | |
| Sandhu *et al.* 2023 [33] | Weekly skill-based learning and education including mindfulness, relaxation, opioid education, consultation and followed by an opioid taper (weekly 10% reduction) | 12 months | Nurse |
| Wartko *et al.* 2023 [35] | CBT-based pain coping skill training: 18 sessions (30 min for each on average), mainly including pain education, goal setting, relaxation skills and motivational interviews, conducted over one year. | 12 months | Primary care provider, nurse, physician assistant |
| Hudak *et al.* 2021 [32] | The MORE protocol: 2-hour weekly training sessions in mindfulness, reappraisal and savoring pleasant events & a 15-min CD-guided mindfulness practice at home and 3-minute breathing before taking opioids | 4 months | Clinical social workers |
| Garland *et al.* 2020 [31] | The MORE protocol: 2-hour weekly training sessions in mindfulness, reappraisal and savoring pleasant events & a 15-min CD-guided mindfulness practice at home and 3-minute breathing before taking opioids | 3 months | Clinical social workers |
| Goodman *et al.* 2018 [39] | Family physician-patient's discussion of pain management guidelines and practice followed by an opioid tapering program (weekly 10% reduction to 25%-50% reduction every few days) | 6 months | Family physician |
| **Multimodalities (n = 3)** | | | |
| Sullivan *et al.* 2017 [34] | A weekly 10% reduction of the initial dose until 30% was reached. Then, a 10% was recalculated based on this dose and then proceeded by 10% of this new dose per week.) & 17 weekly 30-min consultations, CBT-modelled pain self-management, education & a book/CD-guided home practice | 8.5 months | Pain medicine/psychiatry physicians, psychologists and physician assistants |
| Oldfield *et al.* 2018 [38] | Multidisciplinary ORC program including pharmacotherapy, CBT and other modalities (e.g., acupuncture, chiropractic, and yoga) | 6 months | Physicians, pain specialists, pain psychologists and chemical dependency counsellors |
| Seal *et al.* 2019 [37] | Integrated Pain Team: an initial 60-min's biopsychosocial model of pain management (including CBT, mindfulness, acupuncture, chiropractic care, and exercise) & a 30-min follow-up visits | 6 months | Medical provider, psychologist, and pharmacist |

Note: MORE = Mindfulness-Oriented Recovery Enhancement, CBT = Cognitive Behavioral Therapy, ORC = Opioid Reassessment Clinic, GMV = Group Medical Visits

(MD: -37.07 mg; 95%CI: -57.01 to -17.12; $I^2$ = 47.58%; Fig 7). Subgroup analysis of Oldfield et al. (2018) and Seal et al. (2021) [37] using CBT-based interventions showed a significant reduction in opioid dose, compared to those in the control group. (MD: -41.68 mg; 95%CI: -58.47 to -24.89; $I^2$ = 0.00%; Fig 7).

## Opioid cessation

No significant differences were observed in opioid cessation (Odds ratio: 1.52, 95%CI: 0.95 to 2.10; $I^2$ = 0.00%; two trials; Fig 8) between intervention groups and controls. In the two RCTs [33, 34], although the interventions had slight variations, both studies implemented a weekly 10% dose reduction protocol combined with education and consultation as core components of the interventions. In the Sandhu et al. (2023) study [33], it was reported that patients undergoing skill-based learning and education had a significantly higher cessation rate than those receiving usual care (Odds radio [OR]: 5.55; 95% CI 2.80 to 10.99). However, our pooled meta-analysis showed no statistically significant change in opioid discontinuation (OR: 1.10; 95%CI -0.48 to 2.67; Fig 8). Moderate heterogeneity ($I^2$ = 58.59%, 95% CI 0% to 86.2%) was observed, primarily attributed to the small sample size of the Sullivan et al. (2017) study [34].

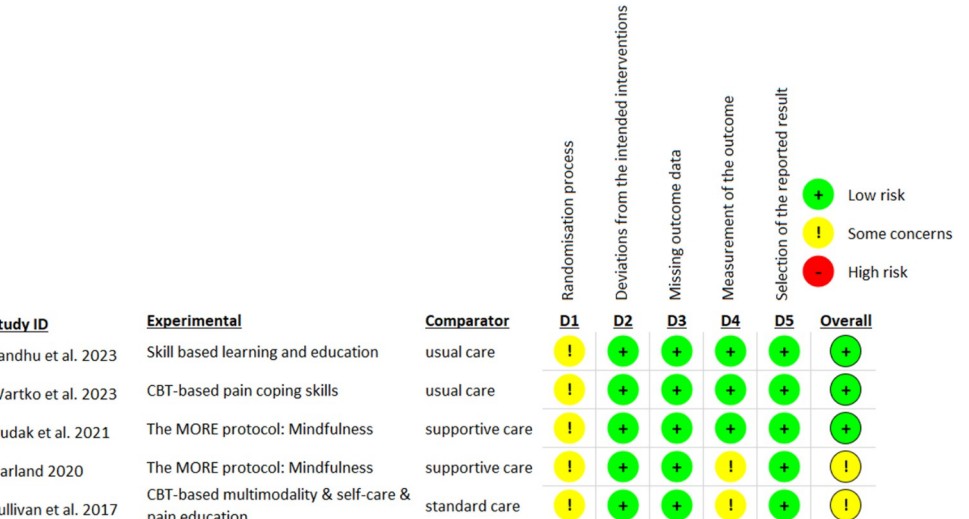

**Fig 2. Risk of Bias of included RCTs.**

## Pain severity

Pain severity changes at different observation time points were reported in three trials [33–35], all utilizing interventions grounded in behavior change and patient education. Due to the variations in pain severity measurements across studies, standard mean differences were used for pooling the estimates. While a decrease in pain severity was observed in all three trials, the between-group difference was small and non-significant (SMD -0.13; 95% CI -0.37 to 0.11, $I^2$ = 33.51%; Fig 9). It is also worth noting that this 0.13-point reduction in pain score might not have meaningful clinical implications.

One cohort study [36] reported the outcome of pain severity improvement. Acupuncture, as employed in the study, exhibited an immediate effect in alleviating pain (a reduction in pain intensity score of MD 1.3 on a scale of 0–10, p<0.01). However, there were no significant differences after 9 months (p = 0.15), indicating acupuncture's short-term effects.

## Adverse events

Adverse events (AEs) were infrequently examined in included studies, with the majority of reported AEs being associated with psychological and nervous system effects. In Wartko et al. (2023)'s study [35], which involved 79 participants receiving CBT-based pain coping skills training plus usual care, six cases of increased pain, one case of withdrawal symptoms, and one case of anxiety were noted. No serious adverse events (SAEs) were documented in this study. In Sandhu et al. (2023) [33], adverse events such as sleep disturbance, suicidal ideation,

**Table 3. Quality assessment outcomes of cohort studies.**

| Author and year | Three domains of the Newcastle-Ottawa Scale | | | Quality |
|---|---|---|---|---|
| | **Selection of participants** | **Comparability of study groups** | **Outcomes** | |
| Montgomery *et al.* 2020 [36] | ★★★ | ★ | ★ | Poor |
| Seal *et al.* 2019 [37] | ★★★★ | ★★ | ★★★ | Good |
| Oldfield *et al.* 2018 [38] | ★★★ | ★★ | ★★★ | Good |
| Goodman *et al.* 2018 [39] | ★★★ | ★ | ★★ | Fair |
| Mehl-Madrona *et al.* 2016 [40] | ★★★ | ★★ | ★★★ | Good |

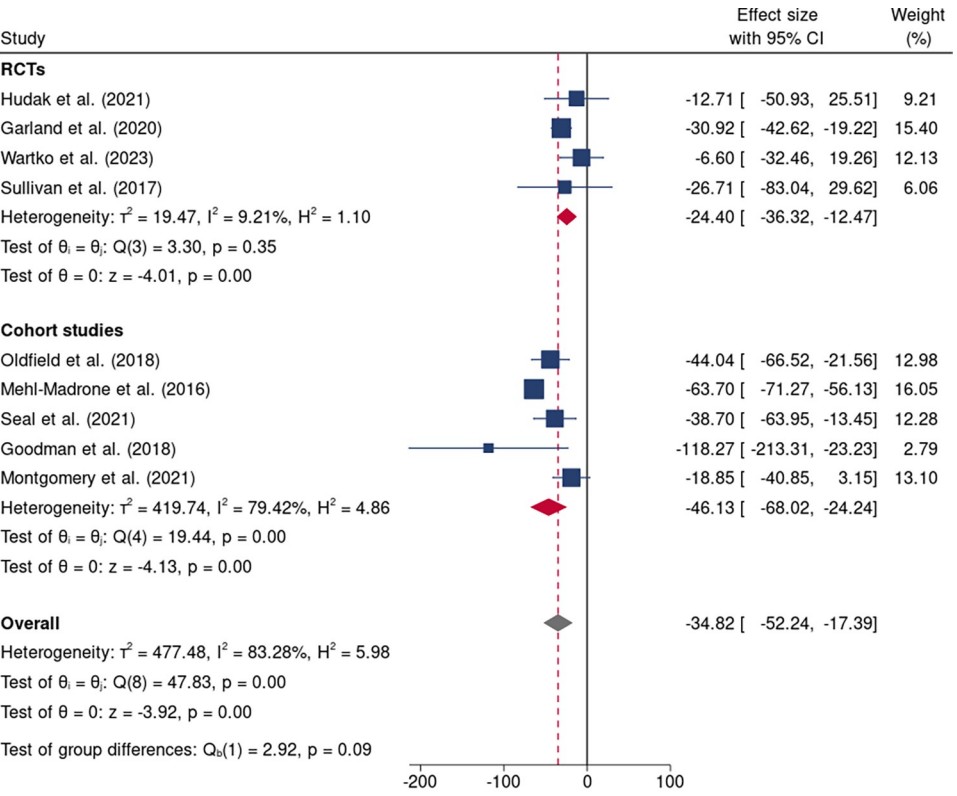

**Fig 3. Forest plot of interventions vs controls in opioid dose reduction in nine studies.**

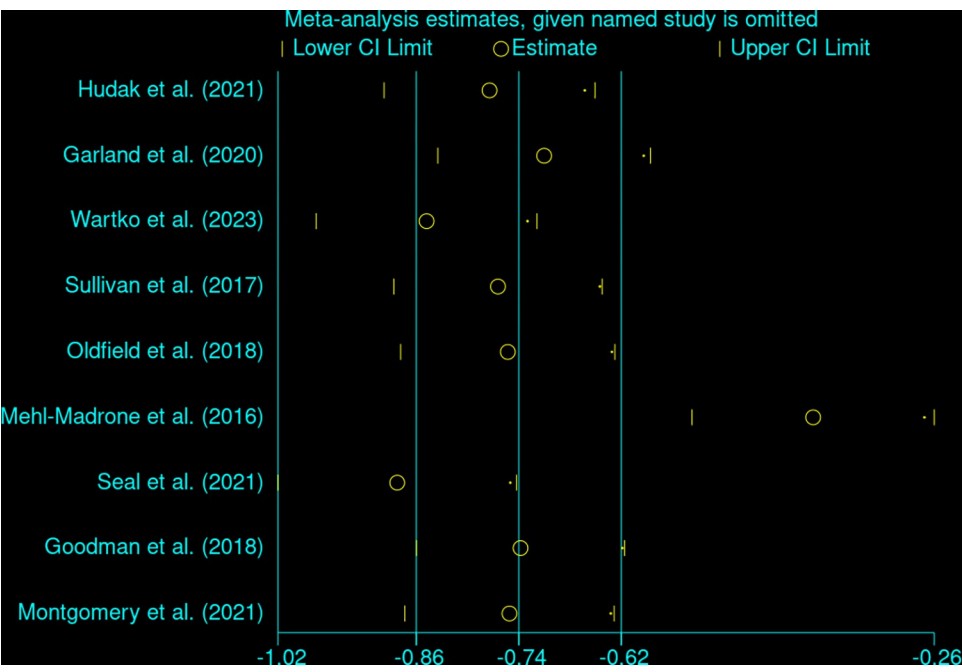

**Fig 4. Sensitivity analysis of included studies.**

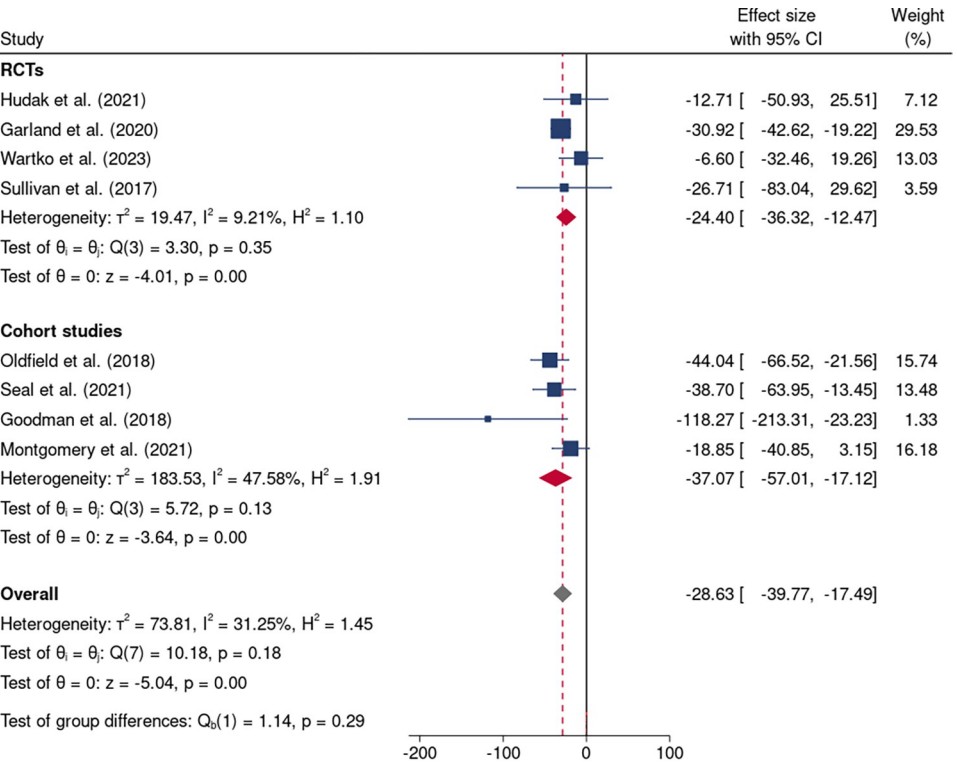

**Fig 5. Forest plot of interventions vs controls in opioid dose reduction in eight studies.**

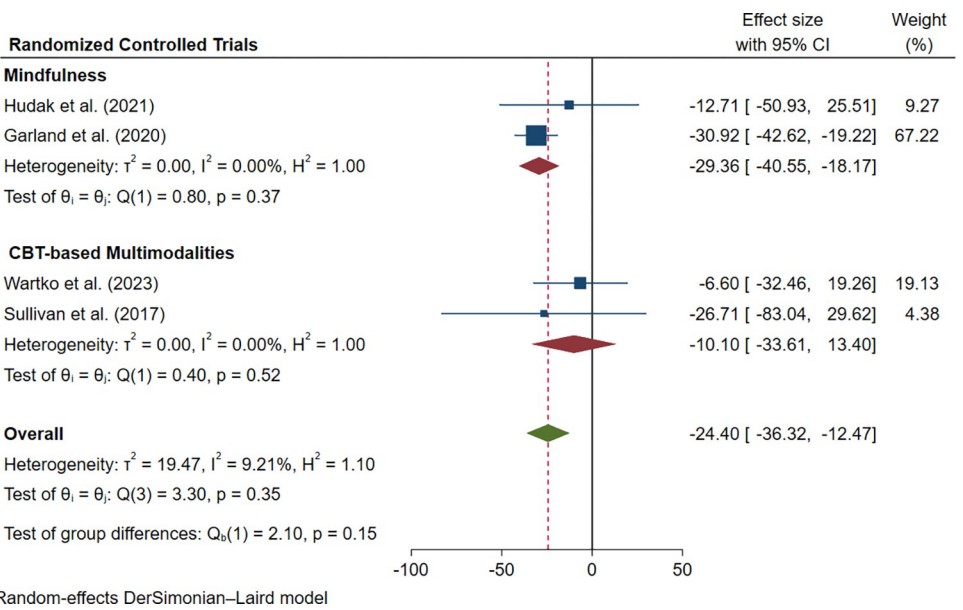

**Fig 6. Forest plot of interventions vs controls in opioid dose reduction in four RCTs.**

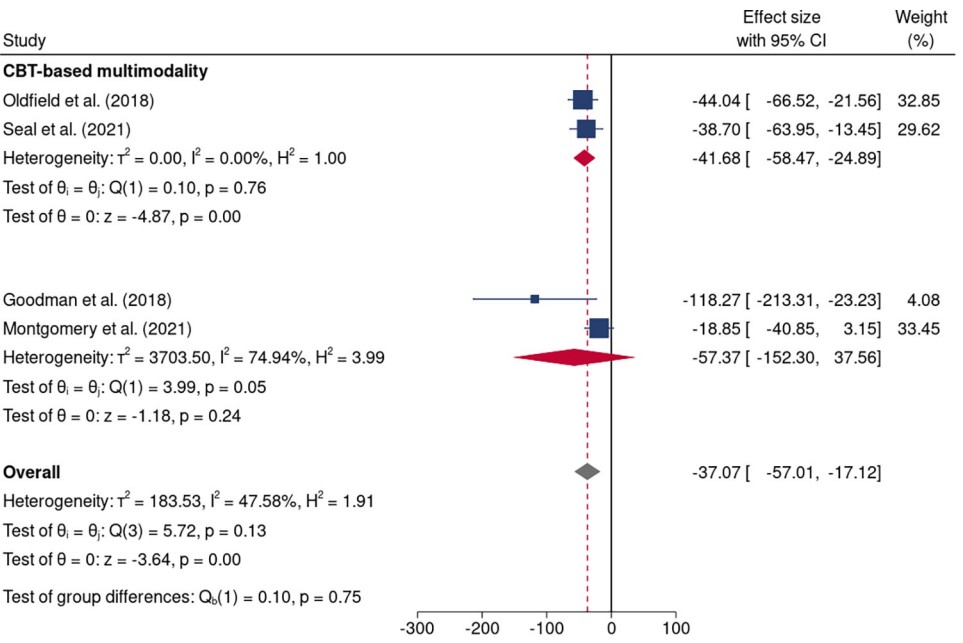

**Fig 7. Forest plot of interventions vs controls in opioid reduction in four cohort studies.**

headache, withdrawal symptoms were reported by 22 of 305 participants (7%) and 8 of 303 participants (3%), in the intervention and usual care groups, respectively. SAEs occurred in 8% (25/305) of the participants in the intervention group and 5% (16/303) in the usual care group, with the most common SAEs being gastrointestinal disorder, metastatic cancer, hospitalization due to increased pain, and unknown cause of death.

## Discussion

### Summary of main findings

In this systematic review and meta-analysis, we examined ten studies (five RCTs and five cohort studies) assessing the effectiveness of opioid reduction interventions for patients with CNCP in primary healthcare settings. Despite methodological differences, both RCTs and observational studies yielded statistically similar results, allowing for the pooling of results

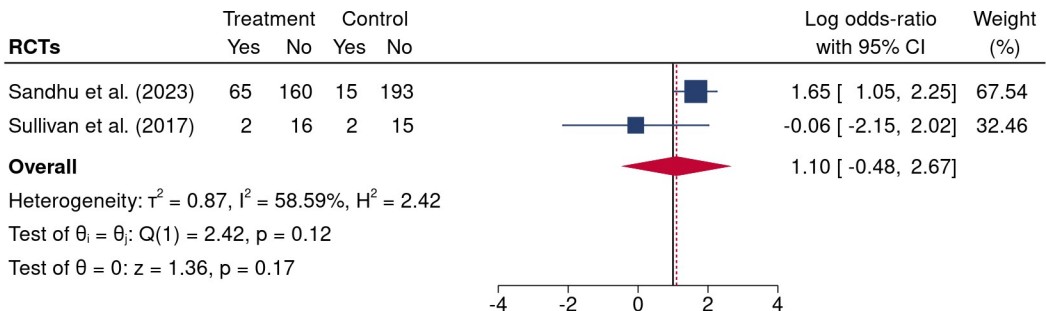

**Fig 8. Forest plot of interventions vs controls in opioid cessation.**

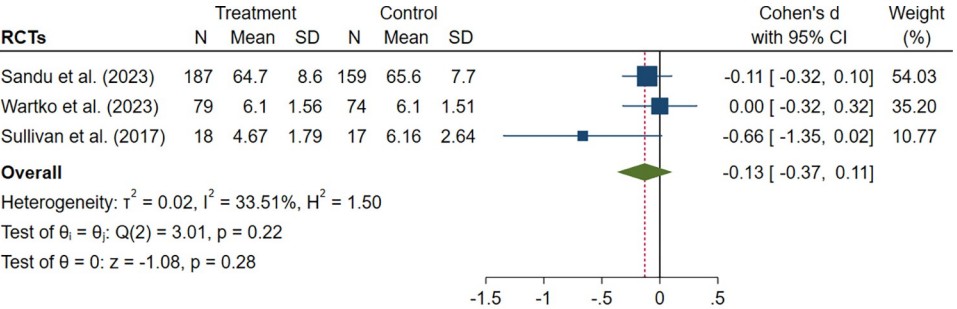

**Fig 9. Forest plot of interventions vs controls in pain severity improvement.**

across the full range of eligible studies. Overall, intervention groups exhibited significant reductions in opioid dosage compared to usual care. Subgroup analyses revealed both mindfulness and CBT-based multimodalities were superior to usual care. No significant differences were observed in opioid cessation or pain severity. Adverse events were infrequently examined, with withdrawal symptoms commonly reported. Medium heterogeneity was observed across all studies for each outcome, likely stemming from variability in interventions, follow-up durations, and healthcare providers involved. Nevertheless, the $I^2$ values for subgroup analyses were low.

## Comparison with existing literature

Our study's conclusions generally aligns with previous systematic reviews [16–19] which indicated that due to insufficient evidence strength and small sample sizes, recommending specific opioid reduction interventions is not feasible. However, our study extends these reviews by providing an updated and comprehensive assessment of the literature, adding one recent good-quality RCT. Supported by subgroup meta-analyses, we elucidated the effectiveness of mindfulness in reducing LTOT. Meanwhile, we highlighted key components such as CBT, patient education, mindfulness, exercise, which may contribute to the successful opioid reduction and thus merit consideration for future intervention development. Importantly, our study exclusively targeted primary care settings, distinguishing it from others that included secondary and tertiary care settings. Although de Kleijn et al. (2021) [17] solely focused on primary care settings, their results were narratively synthesized, and they included studies using simple opioid tapering protocols or interventions targeted at physicians.

Clinical guidelines [11, 12, 41] recommend an initial reduction of opioid dosage by 10% per week, with adjustments to a more tolerable 10% dose reduction per month for high-dose users (> MEDD 120 mg/day) or those on LTOT. In our study, we observed a pooled mean opioid reduction of 28.63 mg/day from a baseline of 84.7 mg/day over a median follow-up period of six months. According to these guidelines, the dosage should have been reduced to 45.01 mg/day over six months. Thus, our observed reduction to 28.63 mg/day appears somewhat aggressive. Despite exceeding the guideline-recommended reduction rate, it did yield a clinically meaningful decrease in opioid dosage without worsening pain severity. It is important to acknowledge that adherence to guideline recommendations does not necessarily guarantee optimal patient outcomes, as individual responses to opioid reduction strategies may vary. In fact, the primary goal of CNCP management is to maintain body function rather than achieve complete pain eradication [42, 43]. Therefore, efforts should prioritize reducing opioids to minimize potential harms associated with LTOT. Meanwhile, although there was infrequent

documentation of the adverse events in the original studies, we did analyze their occurrences (e.g., withdrawal symptoms, increased pain severity, suicides or all-cause mortality). Our findings align with existing evidence [44, 45], suggesting that opioid reduction may lead to these adverse outcomes. Therefore, close monitoring of patient responses to opioid reduction strategies is essential for optimizing treatment outcomes, while mitigating adverse effects.

Our findings highlighted the effectiveness of the MORE protocol, with mindfulness as its core component, in reducing opioid dose for CNCP patients in the short term (3 months). Additionally, meta-analytic results indicated a significant and strong association between mindfulness and pain reduction [46]. In another study [47] conducted by the same author using the MORE protocol, participants reported significantly greater improvements in psychological and physical function outcomes (e.g., general activity, mood, walking ability, normal work, relationships, sleep, and enjoyment of life.) compared to the support group during the three-month follow-up period. In view of this evidence, mindfulness could be considered a potential component in the development of opioid reduction strategies. However, before recommending widespread implementation of this intervention, further validation through replication in a second trial with a longer observation period ($\geq$12 months) within primary care settings is necessary.

Contrary to expectations, CBT-based interventions did not show a significant reduction in opioid dosage in RCTs [34, 35]. This finding aligns with recent evidence from a recent cluster RCT, indicating that although CBT significantly reduced pain severity, it did not result in significant differences in opioid use between groups [43]. The reason for this non-significant result may be twofold. First, the small sample size and variations in interventions across studies may have contributed to the lack of statistical significance. Second, the study conducted by Sullivan et al. (2017) [34] involved high-dose users (207±245 mg/day), suggesting that behavior changes might not be superior to usual care in reducing opioids dosage for this population. However, the pain relief achieved through CBT may contribute to a subsequent decrease in opioid utilization, although this effect may take time to manifest. Two observational studies [37, 38] employing similar CBT-based interventions reported significant reductions in opioid dosage compared to usual care. This inconsistency with findings from RCTs may be attributed to the variations in interventions. Additionally, observational studies may yield more promising results due to unadjusted confounders. Further investigation with long-term follow-up and larger sample sizes is warranted to elucidate the effects of CBT on opioid reduction.

Current clinical guidelines from the UK NICE (National Institute for Health and Care Excellence) [13, 48] and NHS Oxford Hospital [12] advocate for a holistic approach beyond pharmacotherapy alone. Our research findings align with these recommendations, revealing that a combination of cognitive change strategies, acupuncture, exercise, patient education and pharmacotherapies, delivered by a multidisciplinary team were more effective in opioid reduction. Central to this success is the practice of shared decision making between patient and healthcare professionals. The collaboration among diverse healthcare professionals (GPs, nurses, therapists, pharmacists, etc) is crucial, as each contributes unique expertise and insights to the table. Nevertheless, challenges remain in accessing to multimodal cares, particularly for individuals unable to take time off work, those residing in regions with limited services, and individuals from culturally and linguistically diverse communities. Addressing these access barriers is essential to ensuring equitable healthcare delivery.

## Strengths and limitations

To our knowledge, this is the first and most up-to-date systematic review and meta-analysis, to evaluate the effectiveness of interventions on opioid dose reduction and pain outcomes for

patients with CNCP, with a particular emphasis on primary healthcare settings. By conducting a comprehensive literature search and including recently conducted studies in the UK, this research holds informative value for the potential implementation of interventions outside the US primary care settings. Moreover, several key subgroup analyses were conducted to identify which intervention (components) are effective in achieving opioid reduction purpose.

Our study also had several limitations. First, the relatively small number of included participants, short follow-up periods, and lack of blinding to interventions may have introduced biases and impacted the overall quality of the included studies. Second, data in observational studies were sourced from secondary databases or electronic medical records, which could cause information bias. Third, funnel plots and the Egger's test were not performed due to the small number of studies included in the meta-analyses Fourth, the majority of the studies were conducted on white populations in the US, likely influenced by the opioid crisis in that country, thus limited the generalizability of our research findings to other ethnicities and regions. Fifth, the oversight of unprescribed use of opioids may overestimate the effectiveness of opioid reduction strategies, as patients might still be obtaining opioids from unauthorized sources. Conversely, it might also underestimate their effectiveness, since total opioid consumption could be higher than reported. Lastly, data was insufficient to support meta-analyses of quality of life or function.

## Conclusion

Prior studies have not identified superior opioid reduction interventions for patients with CNCP in primary care. Our study highlighted those multidisciplinary strategies, incorporating CBT, pain education, mindfulness, exercise and acupuncture, which have shown promise in reducing opioids and improving pain outcomes for this population. Nevertheless, robust evidence is limited due to sub-optimally designed studies, short follow-up periods, and a lack of QoL-related outcomes. Future research should focus on identifying the effective components of these interventions and identifying which CNCP sub-populations benefit most. Additionally, it is essential to assess the cost-effectiveness and patient acceptability of these strategies, before their implementation in primary care contexts.

## Supporting information

**S1 Appendix. Search strategies on the five medical databases.**
(DOCX)

**S2 Appendix. PRISMA 2009 checklist.**
(DOC)

## Author Contributions

**Conceptualization:** Qian Cai, Teng-Chou Chen, Li-Chia Chen, Evangelos Kontopantelis.

**Data curation:** Qian Cai, Christos Grigoroglou, Teng-Chou Chen, Evangelos Kontopantelis.

**Formal analysis:** Qian Cai, Evangelos Kontopantelis.

**Investigation:** Qian Cai, Christos Grigoroglou, Teng-Chou Chen.

**Methodology:** Qian Cai, Evangelos Kontopantelis.

**Project administration:** Qian Cai.

**Supervision:** Christos Grigoroglou, Thomas Allen, Li-Chia Chen, Evangelos Kontopantelis.

**Validation:** Qian Cai, Christos Grigoroglou, Teng-Chou Chen.

**Visualization:** Qian Cai.

**Writing – original draft:** Qian Cai.

**Writing – review & editing:** Qian Cai, Christos Grigoroglou, Thomas Allen, Teng-Chou Chen, Li-Chia Chen, Evangelos Kontopantelis.

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
