## [Decision Letter · Decision Letter 0]

12 Jun 2024

PONE-D-24-09355Interventions to reduce opioid use for patients with chronic non-cancer pain in primary care settings: a systematic review and meta-analysisPLOS ONE

Dear Dr. Cai,

Thank you for submitting your manuscript to PLOS ONE. After careful consideration, we feel that it has merit but does not fully meet PLOS ONE’s publication criteria as it currently stands. Therefore, we invite you to submit a revised version of the manuscript that addresses the points raised during the review process.

As you know based on our communication, we did have trouble securing reviewers for this manuscript. Fortunately, we secured one thoughtful and highly skilled reviewer who provided detailed feedback. I encourage you to address all of their comments as I believe they will lead to a much improved manuscript. If you choose to revise and resubmit, we will try to obtain additional feedback from other peers.

We look forward to receiving your revised manuscript.

Kind regards,

Ethan Moitra

Academic Editor

PLOS ONE

Journal Requirements:

Reviewers' comments:

Reviewer's Responses to Questions

**Comments to the Author**

1. Is the manuscript technically sound, and do the data support the conclusions?

Reviewer #1: Partly

2. Has the statistical analysis been performed appropriately and rigorously? 

Reviewer #1: Yes

3. Have the authors made all data underlying the findings in their manuscript fully available?

Reviewer #1: Yes

4. Is the manuscript presented in an intelligible fashion and written in standard English?

Reviewer #1: Yes

5. Review Comments to the Author

Reviewer #1: Thank you for the chance to review this article, which I think accurately represents the limited but promising data around behavioral interventions which may have benefits for patients with CNCP. The following suggestions may help strengthen this paper further:

Introduction:

Lines 96-99:This is particularly common for studies using pharmacological substitution, where the primary aim was to address opioid withdrawal symptoms rather than reducing opioid dosage, despite achieving a decrease in opioid dosage or an improvement in pain severity

Add supporting citations. I found this surprising given my own awareness of the literature, which does not tend to focus on opioid withdrawal to the exclusion of dose reduction.

Results:

Lines 231-233: Rather than say "most" studies did not report type of CNCP, mention exact numbers, and the exception. Verify this is correct- as many studies do breakdown neuropathic pain vs msk pain, etc.

Were patients with substance use disorders, e.g. opioid use disorders, excluded? Was unprescribed use of opioids examined in some way? Include these in limitations later on as appropriate.

Table 1: Consider including what type of CNCP was investigated, if applicable.

Please justify inclusion of the study of medical cannabis if other potential analgesics were excluded in favor of behavioral interventions? The impact of cannabis on pain is a complex topic that seems beyond the scope of this study. I would consider excluding this study from analysis entirely, given that it is the only one involving cannabis, and also has a small sample size.

Was "usual care" defined in these studies, and if so, how?

Some interpretation is included in the Results section which might be more appropriate for the Discussion, e.g.:

Lines 284-85: Furthermore, a multivariable meta-regression analysis was conducted, revealing that heterogeneity could be partially explained by differences in the longest follow-up time across studies (p=0.014).

Lines 337-38:It is worth noting that this 0.13-point reduction in pain score might not have meaningful clinical implications.

Discussion:

Lines 381: Consider changing to "Comparison with existing literature"

Lines 382-383:Our study conclusion generally aligns with previous systematic reviews [16-19] that the strength of evidence was insufficient to draw conclusions.

This line is a bit confusing, as you are clearly drawing some conclusions and reporting them in this paper. Re-phrase- do you mean the sample sizes and quality of evidence were so low that your results do not challenge previous studies?

Lines 397-99:In our study, we observed a pooled mean opioid reduction of 24.88 mg/day from a baseline of 87.4 mg/day. Although this reduction was aggressive and exceeded the guideline-recommended reduction rate...

Reiterate the average and/or median timeframe over which this reduction occurred. Without this information, the following statement that "reduction was aggressive" doesn't make sense.

Lines 403-404: In fact, the primary goal of CNCP management is to maintain body function rather than achieve complete pain eradiation.

Add citation for this statement

Lines 406-408: in this study, our findings align with existing evidence [40, 41], suggesting that opioid reduction may lead to withdrawal symptoms, increased pain severity, suicides or all-cause mortality.

Clarify: since these were not reported in the studies you examined, do you mean that they may be of concern in larger populations followed for longer periods of time?

Lines 442-55: Cannabis section- examining the effects of cannabis on opioid utilization and pain seems beyond the scope of this paper, especially given the complexity of this issue, including variable effects of different cannabis products and cannabinoids on pain. I would consider excluding or limiting this section.

Lines 493-97:Although according to prior studies, no specific interventions can be recommended over one another, multidisciplinary opioid reduction strategies, incorporating components such as CBT, pain education, mindfulness, exercise and acupuncture, have demonstrated effectiveness and tolerability in reducing opioid doses and improving pain severity among patients with CNCP in primary care settings.

The way this is phrased is a bit confusing. Please rephrase and make more concise.

6. PLOS authors have the option to publish the peer review history of their article (what does this mean?). If published, this will include your full peer review and any attached files.

Reviewer #1: No

---

## [Author Response · Author response to Decision Letter 0]

29 Jun 2024

Response to academic editor and reviewer(s)

Journal: PLOS ONE

Title: Interventions to reduce opioid use for patients with chronic non-cancer pain in primary care settings: a systematic review and meta-analysis

Paper ID: PONE-D-24-09355

Dear Ethan,

We appreciate your time and effort in securing reviewers for our study. Both your and the reviewer’s comments are helpful in improving our work. According to your suggestions, we have revised the manuscript. All issues are addressed point by point as follow.

Revision suggestions from Ethan (Academic Editor).

 These have been done.

2. We didn’t make any changes to the financial disclosure.

3. If applicable, we recommend that you deposit your laboratory protocols in protocols.io to enhance the reproducibility of your results. Protocols.io assigns your protocol its own identifier (DOI) so that it can be cited independently in the future. 

For instructions see: https://journals.plos.org/plosone/s/submission-guidelines#loc-laboratory-protocols. 

Additionally, PLOS ONE offers an option for publishing peer-reviewed Lab Protocol articles, which describe protocols hosted on protocols.io. Read more information on sharing protocols at https://plos.org/protocols?utm_medium=editorial-email&utm_source=authorletters&utm_campaign=protocols.

The research protocol is available with PROSPERO (CRD42022338458) and this is clearly stated in the manuscript.

Journal Requirements:

The title page has been revised per instructions (i.e., corresponding authors’ phone number and physical address have been removed; the initial of the corresponding author’s name has been put into parentheses after the email). 

The manuscript format has been revised to meet PLOS ONE’s style requirements (i.e., information including Contributors and Funding has been removed, and supporting information has been added right below References) 

Captions of Supporting Information files have been included at the end of the manuscript right after the References

We have updated the reference list using the Vancouver style. 

Comments from Reviewer #1:

1. Introduction:

Lines 96-99:This is particularly common for studies using pharmacological substitution, where the primary aim was to address opioid withdrawal symptoms rather than reducing opioid dosage, despite achieving a decrease in opioid dosage or an improvement in pain severity

Add supporting citations. I found this surprising given my own awareness of the literature, which does not tend to focus on opioid withdrawal to the exclusion of dose reduction.

Thank you for this comment. We carefully reviewed the studies included in prior published systematic reviews again, and found three articles using pharmacological interventions, but their primary purpose was not to reduce opioids, despite dose reduction outcomes occurred during the process. To clarify, we revised the statement in the Introduction section, in lines 92-96, as follows. (unless otherwise stated, all line numbers in this letter pertain to the revised manuscript without track changes)

“Some prior systematic reviews also included potentially relevant studies. However, those studies focused on managing related symptoms, such as withdrawal symptoms [20], opioid overuse headache [21], or improving adherence to antidepressants to increase the likelihood of opioid cessation [22], rather than directly reducing opioid use, despite a decrease in opioid dosage or an improvement in pain severity was achieved.” 

Below, reference 20-22 have been added in the reference list.

20. Hooten WM, Warner DO. Varenicline for opioid withdrawal in patients with chronic pain: a randomized, single-blinded, placebo controlled pilot trial. Addictive Behaviors. 2015;42:69-72.

21. Johnson JL, Kwok YH, Sumracki NM, Swift JE, Hutchinson MR, Johnson K, et al. Glial attenuation with ibudilast in the treatment of medication overuse headache: a double‐blind, randomized, placebo‐controlled pilot trial of efficacy and safety. Headache: The Journal of Head and Face Pain. 2015;55(9):1192-208.

22. Scherrer JF, Salas J, Sullivan MD, Ahmedani BK, Copeland LA, Bucholz KK, et al. Impact of adherence to antidepressants on long-term prescription opioid use cessation. The British Journal of Psychiatry. 2018;212(2):103-11.

2. Results:

Lines 231-233: Rather than say "most" studies did not report type of CNCP, mention exact numbers, and the exception. Verify this is correct- as many studies do breakdown neuropathic pain vs msk pain, etc.

Thank you for raising this. We have revised the expression from "most studies" to "None of the studies specified the CNCP conditions." This adjustment reflects the exclusion of the study involving a medical cannabis substitution program, which was the only study explicitly mentioning chronic musculoskeletal pain conditions.

3. Were patients with substance use disorders, e.g. opioid use disorders, excluded? Was unprescribed use of opioids examined in some way? Include these in limitations later on as appropriate.

Indeed, cohorts of patients with substance use disorders were excluded. Thank you for the suggestion. To clarify the inclusion/exclusion criteria, we added below sentence in the Eligibility criteria section, lines 118-121.

“Studies exclusively focused on acute pain, cancer pain, surgical pain or palliative care were excluded. Pregnant or breastfeeding women, non-human participants and patients with substance use disorders (e.g. opioid use disorders) were not our target study population and were thus excluded.”

We understand that some opioids can be illegally acquired without prescriptions from healthcare providers. Unfortunately, none of the included studies examined this. According to your suggestion, we added the below, explaining this limitation in the Discussion section, lines 475-479.

“Fifth, the oversight of unprescribed use of opioids may overestimate the effectiveness of opioid reduction strategies, as patients might still be obtaining opioids from unauthorized sources. Conversely, it might also underestimate their effectiveness, since total opioid consumption could be higher than reported.”

4. Table 1: Consider including what type of CNCP was investigated, if applicable.

Thank you for this comment. We fully understand the importance of specifying the type of CNCP to better identify which patients are more likely to benefit from opioid reduction. Unfortunately, only one of the included studies specified musculoskeletal pain as the CNCP condition, and this study was excluded due to its cannabis intervention. The remaining studies only used the general term -- CNCP. 

Considering this, we added the following sentence to the Conclusion section, lines 488-490. “Future research should focus on identifying the effective components of these interventions and specifying which CNCP conditions benefit most.” 

5. Please justify inclusion of the study of medical cannabis if other potential analgesics were excluded in favor of behavioral interventions? The impact of cannabis on pain is a complex topic that seems beyond the scope of this study. I would consider excluding this study from analysis entirely, given that it is the only one involving cannabis, and also has a small sample size.

Thank you for this suggestion. After careful consideration, we have decided to exclude this study. Accordingly, analyses and discussions related to cannabis and pharmacological interventions were revised. 

6. Was "usual care" defined in these studies, and if so, how?

We acknowledge the importance of having a clearly defined control group. Unfortunately, the included studies did not specify the components of usual care, which prevented us from providing a detailed description of it. To clarify this situation, we have added the following sentence to lines 252-253:

 “The most common comparator was usual care, which unfortunately was not clearly defined in the included studies”

7. Some interpretation is included in the Results section which might be more appropriate for the Discussion, e.g.:

Lines 284-85: Furthermore, a multivariable meta-regression analysis was conducted, revealing that heterogeneity could be partially explained by differences in the longest follow-up time across studies (p=0.014).

Lines 337-38:It is worth noting that this 0.13-point reduction in pain score might not have meaningful clinical implications.

Thank you for your comment. We understand that traditionally, interpretations are placed in the Discussion section. However, we believe that by presenting the meta-regression findings within the Results section, we ensure that readers can directly see how differences in follow-up times contribute to the observed heterogeneity, thereby providing a more comprehensive understanding of the results. We briefly mention the interpretation of clinical implications of the -0.13 change in pain severity in the results section, so it is immediately accessible to readers. We hope that this is acceptable.

8. Discussion:

Lines 381: Consider changing to "Comparison with existing literature"

The heading has been changed into “Comparison with existing literature” from “Compare with existing literature”.

9. Lines 382-383:Our study conclusion generally aligns with previous systematic reviews [16-19] that the strength of evidence was insufficient to draw conclusions.

This line is a bit confusing, as you are clearly drawing some conclusions and reporting them in this paper. Re-phrase- do you mean the sample sizes and quality of evidence were so low that your results do not challenge previous studies?

Thank you for pointing out this. We revised to make this sentence clearer. See lines 378-380.

“Our study’s conclusions generally align with previous systematic reviews [16-19], which indicate that due to insufficient evidence strength and small sample sizes, recommending specific opioid reduction strategies is not feasible.”

10. Lines 397-99:In our study, we observed a pooled mean opioid reduction of 24.88 mg/day from a baseline of 87.4 mg/day. Although this reduction was aggressive and exceeded the guideline-recommended reduction rate...

Reiterate the average and/or median timeframe over which this reduction occurred. Without this information, the following statement that "reduction was aggressive" doesn't make sense.

Thank you very much. This is really a good point. We calculated a median follow-up duration of 6 months, during which the pooled mean opioid reduction was 28.63 mg/day from a baseline of 84.70 mg/day over a period of six months. According to the 2016 CDC guideline, a 10% dose reduction per month is more tolerable for patients on long-term opioids, suggesting a reduction to 45.01 mg/day over six months. Thus, our observed reduction to 28.63 mg/day appears somewhat aggressive.

We have made some revisions to the Comparison with existing literature section, in lines 392-400, as follows:

“Clinical guidelines [11, 12, 42] recommend an initial reduction of opioid dosage by 10% per week, with adjustments to a more tolerable 10% dose reduction per month for high-dose users (> MEDD 120 mg/day) or those on LTOT. In our study, we observed a pooled mean opioid reduction of 28.63 mg/day from a baseline of 84.70 mg/day over a median follow-up period of six months. According to these guidelines, the dosage should have been reduced to 45.01 mg/day over six months. Thus, our observed reduction to 28.63 mg/day appears somewhat aggressive. Despite exceeding the guideline-recommended reduction rate, it did yield a clinically meaningful decrease in opioid dosage without worsening pain severity.”

11. Lines 403-404: In fact, the primary goal of CNCP management is to maintain body function rather than achieve complete pain eradiation.

Add citation for this statement

Thank you for the reminder. We have added below two references to support this statement.

“In fact, the primary goal of CNCP management is to maintain body function rather than achieve complete pain eradiation [42, 43].”

42. Ashburn MA, Staats PS. Management of chronic pain. The lancet. 1999;353(9167):1865-9.

43. Roditi D, Robinson ME. The role of psychological interventions in the management of patients with chronic pain. Psychology research and behavior management. 2011:41-9.

12. Lines 406-408: in this study, our findings align with existing evidence [40, 41], suggesting that opioid reduction may lead to withdrawal symptoms, increased pain severity, suicides or all-cause mortality.

Clarify: since these were not reported in the studies you examined, do you mean that they may be of concern in larger populations followed for longer periods of time?

We apologize for not making the statement clear. Actually, withdrawal symptoms, increased pain severity, suicide and death were examined by the included studies and we reported the relevant proportions in the Adverse events section. To clarify, we have made the below revisions to the Comparison with existing literature in lines 405-411.

“Meanwhile, although there was infrequent documentation of the adverse events in the original studies, we did analyze their occurrences (e.g. withdrawal symptoms, increased pain severity, suicides, or all-cause mortality). Our findings align with existing evidence [44, 45], suggesting that opioid reduction may lead to these adverse outcomes. Therefore, close monitoring of patient responses to opioid reduction strategies is essential for optimizing treatment outcomes, while mitigating adverse effects.”

13. Lines 442-55: Cannabis section- examining the effects of cannabis on opioid utilization and pain seems beyond the scope of this paper, especially given the complexity of this issue, including variable effects of different cannabis products and cannabinoids on pain. I would consider excluding or limiting this section.

Thank you for this suggestion. Since the study involving medical cannabis has been removed, we have also deleted the related discussions accordingly as below: 

Pharmacological interventions (cannabis-assisted opioid substitution) showed promising result, 

---

## [Editor Report · Decision Letter 1]

23 Jul 2024

Interventions to reduce opioid use for patients with chronic non-cancer pain in primary care settings: a systematic review and meta-analysis

PONE-D-24-09355R1

Dear Dr. Cai,

We’re pleased to inform you that your manuscript has been judged scientifically suitable for publication and will be formally accepted for publication once it meets all outstanding technical requirements.

Kind regards,

Ethan Moitra

Academic Editor

PLOS ONE
---

## [Editor Report · Acceptance letter]

25 Jul 2024

PONE-D-24-09355R1 

PLOS ONE

Dear Dr. Cai, 

I'm pleased to inform you that your manuscript has been deemed suitable for publication in PLOS ONE. Congratulations! Your manuscript is now being handed over to our production team.

Kind regards, 

on behalf of

Dr. Ethan Moitra 

Academic Editor

PLOS ONE